# Mushroom body subsets encode CREB2-dependent water-reward long-term memory in *Drosophila*

**Wang-Pao Lee**[1], **Meng-Hsuan Chiang**[1], **Li-Yun Chang**[1], **Jhen-Yi Lee**[2], **Ya-Lun Tsai**[1], **Tai-Hsiang Chiu**[1], **Hsueh-Cheng Chiang**[3], **Tsai-Feng Fu**[4], **Tony Wu**[5], **Chia-Lin Wu**[1,5,6]*

**1** Graduate Institute of Biomedical Sciences, College of Medicine, Chang Gung University, Taiwan, **2** School of Medicine, College of Medicine, Chang Gung University, Taiwan, **3** Department of Pharmacology, National Cheng-Kung University, Taiwan, **4** Department of Applied Chemistry, National Chi Nan University, Taiwan, **5** Department of Neurology, Chang Gung Memorial Hospital, Taiwan, **6** Department of Biochemistry, College of Medicine, Chang Gung University, Taiwan

* clwu@mail.cgu.edu.tw

**Data Availability Statement:** All relevant data are within the manuscript and its Supporting Information files.

## Abstract

Long-term memory (LTM) formation depends on the conversed cAMP response element-binding protein (CREB)-dependent gene transcription followed by *de novo* protein synthesis. Thirsty fruit flies can be trained to associate an odor with water reward to form water-reward LTM (wLTM), which can last for over 24 hours without a significant decline. The role of *de novo* protein synthesis and CREB-regulated gene expression changes in neural circuits that contribute to wLTM remains unclear. Here, we show that acute inhibition of protein synthesis in the mushroom body (MB) αβ or γ neurons during memory formation using a cold-sensitive ribosome-inactivating toxin disrupts wLTM. Furthermore, adult stage-specific expression of *dCREB2b* in αβ or γ neurons also disrupts wLTM. The MB αβ and γ neurons can be further classified into five different neuronal subsets including αβ core, αβ surface, αβ posterior, γ main, and γ dorsal. We observed that the neurotransmission from αβ surface and γ dorsal neuron subsets is required for wLTM retrieval, whereas the αβ core, αβ posterior, and γ main are dispensable. Adult stage-specific expression of *dCREB2b* in αβ surface and γ dorsal neurons inhibits wLTM formation. *In vivo* calcium imaging revealed that αβ surface and γ dorsal neurons form wLTM traces with different dynamic properties, and these memory traces are abolished by *dCREB2b* expression. Our results suggest that a small population of neurons within the MB circuits support long-term storage of water-reward memory in *Drosophila*.

## Author summary

Unlike short-term memory (STM), the formation of long-term memory (LTM) requires *de novo* protein synthesis and CREB-mediated gene transcription in many animals. To date, the mechanism underlying LTM formation remains poorly understood. Thirsty fruit flies can be trained to associate an odor with water to form a water-reward LTM (wLTM),

**Funding:** This work was supported by grants from the Ministry of Science and Technology (106-2311-B-182-004-MY3 and 109-2326-B-182-001-MY3) to CLW, Chang Gung Memorial Hospital (CMRPD1G0341-3, CMRPD1K0311-2, and BMRPC75) to CLW, and Higher Education Sprout Project funded by the Ministry of Science and Technology and Ministry of Education to CLW. The funders had no role in study design, data collection and analysis, decision to publish, or preparation of the manuscript.

**Competing interests:** The authors have declared that no competing interests exist.

which requires *de novo* protein synthesis and *dCREB2* activity. In this study, we found that *dCREB2* activity in the mushroom body (MB) αβ surface and γ dorsal neuron subsets is essential for wLTM formation. Neurotransmission from αβ surface and γ dorsal neurons is specifically required for retrieval, but not for acquisition or consolidation of wLTM. Moreover, wLTM traces are formed in the αβ surface and γ dorsal neurons with different neural dynamics, which require normal *dCREB2* functions. These findings highlight that *dCREB2*-dependent wLTM is located within a specific brain circuitry in fruit flies.

## Introduction

Long-term memory (LTM) requires hours to transform labile memory to the long-lasting form of memory and needs *de novo* protein synthesis to support the synaptic morphology changes [1, 2, 3]. Moreover, in many species, LTM formation requires cAMP-response element-binding protein (CREB)-mediated gene transcription and protein synthesis [4–7]. CREB is a leucine-zipper transcription factor, which is evolutionarily conserved across species. The basic region of the leucine-zipper proteins binds to cAMP-response element (CRE) sites and regulates downstream gene expression. CREB knockout mice have been reported to have a defect in LTM although the initial learning and short-term memory (STM) is normal [8]. In *Drosophila*, both shock-punitive and sugar-reward LTMs require CREB2 activity, and ectopic expression of CREB2 repressor (*dCREB2b*) disrupts olfactory LTMs but not STMs [6, 9]. In *Aplysia*, microinjection of the CRE sequence into the nucleus of sensory neurons selectively blocks long-term facilitation without affecting the short-term facilitation [10].

Thirsty fruit flies can be trained to associate a specific odor (a conditioned stimulus, CS) with water (an unconditioned stimulus, US) to form water-reward memory, which can last for over 24 hours [1, 11]. *De novo* protein synthesis and several LTM specific genes are required for this process, suggesting that the long-lasting water memory is an LTM and not STM [1]. The dendritic regions of the mushroom body (MB), traditionally viewed as the olfactory memory center known as calyx, receive olfactory input from the antenna lobe via the projection neurons, and this olfactory information is transformed into sparse neural codes in the MB [12, 13]. The specific dopaminergic protocerebral anterior medial (PAM) neurons, called PAM-β′1 convey the water-rewarding inputs to the MB in thirsty flies [1]. Therefore, the association between an odor and water-reward is established in the MB, which is finally transformed into water-reward LTM (wLTM) [1].

The MB is a paired neuropil structure comprising approximately 2000 Kenyon cells in each brain hemisphere [14]. The Kenyon cells can be divided into αβ, γ, and α′β′ neurons according to their axonal distributions. Here, we report that genetic inactivation of protein synthesis by expressing the activated cold-sensitive *ricin* (RICIN^CS) during memory formation in αβ or γ neurons disrupts wLTM. Furthermore, adult stage-specific expression of *dCREB2b* in αβ or γ neurons also disrupts wLTM. It has been shown that neurotransmission from αβ and γ neurons is required for wLTM retrieval [1]. The αβ and γ neurons can be further classified into five different subsets including αβ core, αβ surface, αβ posterior, γ main, and γ dorsal [15]. Our results showed that the neurotransmission from αβ surface and γ dorsal is required for wLTM retrieval, whereas the αβ core, αβ posterior, and γ main are dispensable. Adult stage-specific expression of *dCREB2b* in αβ surface or γ dorsal also disrupts wLTM. Finally, we observed an increased *in vivo* calcium response in αβ surface and a decreased calcium response in γ dorsal in response to the training odor at 24-hour post-training in thirsty flies. These

training-induced calcium response changes to odors were abolished by *dCREB2b* expression, suggesting that wLTM traces are formed in these MB circuits.

## Results

### *De novo* protein synthesis and CREB2 activity in αβ and γ neurons are critical for wLTM

It has been shown that both shock-punitive and sugar-reward LTMs require *de novo* protein synthesis in *Drosophila* [3, 9]. Our previous study also suggests that *de novo* protein synthesis is essential for wLTM formation [1]. To investigate whether protein synthesis in MB neurons is critical for wLTM formation, we genetically inactivated the protein synthesis in MB neurons by expressing RICIN$^{CS}$, a toxin that inhibits the eukaryotic ribosomes by cleaving the N-glycosidic bond of 28S rRNA [16, 17]. All the flies were kept at 18˚C until eclosion to avoid the developmental effects of RICIN$^{CS}$ and water-reward conditioning was performed at 18˚C. The trained flies were then transferred to 30˚C for 12 hours right after the training, and moved back to 18˚C for another 12 hours and tested at 18˚C. We observed that genetic inactivation of protein synthesis in MB neurons during memory formation disrupts wLTM (Fig 1A and S1 Fig). Furthermore, protein synthesis in MB γ and αβ neurons was found to be essential for wLTM formation, whereas it was dispensable in α′β′ neurons (Fig 1B–1G and S1 Fig). Since LTM formation is a CREB-dependent process that requires gene transcription and *de novo* protein synthesis, it prompted us to ask that CREB activity in which MB neurons are critical for wLTM. It has been shown that constitutive *dCREB2b* expression in MB neurons causes a significant neuroanatomical damage in the MB [18]. We therefore used *tub-GAL80$^{ts}$* to limit *UAS-dCREB2b* expression to whole MB neurons (*OK107-GAL4*), γ neurons (*R16A06-GAL4*), αβ neurons (*C739-GAL4*), or α′β′ neurons (*VT30604-GAL4*) in adult flies to eliminate these developmental defects. Flies were kept at 18˚C until eclosion to avoid *dCREB2b* expression during the developmental stage. After eclosion, adult flies were transferred to 30˚C for *dCREB2b* transgene expression for three days. Water-reward conditioning was performed at 30˚C and the trained flies were kept at 30˚C for 8 hours, and then moved back to 18˚C for 16 hours and tested. Flies in the control group were kept at 18˚C throughout the experiment. We found that the adult stage-specific expression of *dCREB2b* in γ or αβ neurons disrupted the wLTM, whereas *dCREB2b* expression in α′β′ neurons did not affect the wLTM (Fig 2 and S1 Fig).

### wLTM retrieval requires neurotransmission from αβ surface and γ dorsal neuron subsets

The *de novo* protein synthesis and CREB2 activity in αβ and γ neuron subsets were found to be essential for wLTM formation (Figs 1 and 2). This led us to investigate whether the neurotransmission from specific αβ and γ neuron subsets plays a crucial role in wLTM. It has been shown that wLTM retrieval requires neurotransmission from αβ and γ neurons [1]. The αβ and γ neurons can be further classified into αβ core, αβ surface, αβ posterior, γ main, and γ dorsal subsets [15]. We analyzed the role of each αβ and γ neuron subsets in wLTM retrieval. The *GAL4* labelled subsets of αβ and γ neurons were identified including αβ posterior (labeled by *VT14429-GAL4* & *VT24615-GAL4*; Fig 3A and 3C), αβ core (labeled by *VT0841-GAL4* & *VT8347-GAL4;* Fig 3E and 3G), αβ surface (labeled by *VT20803-GAL4* & *VT21845-GAL4*; Fig 3I and 3K), γ main (labeled by *R64C08-GAL4*; Fig 4A), and γ dorsal (*R93G04-GAL4* & *MB607B-GAL4*; Fig 4C and 4E). We genetically expressed the temperature-sensitive *shibire* (*shi$^{ts}$*) transgene in individual αβ and γ neuron subset via these specific GAL4 lines. The

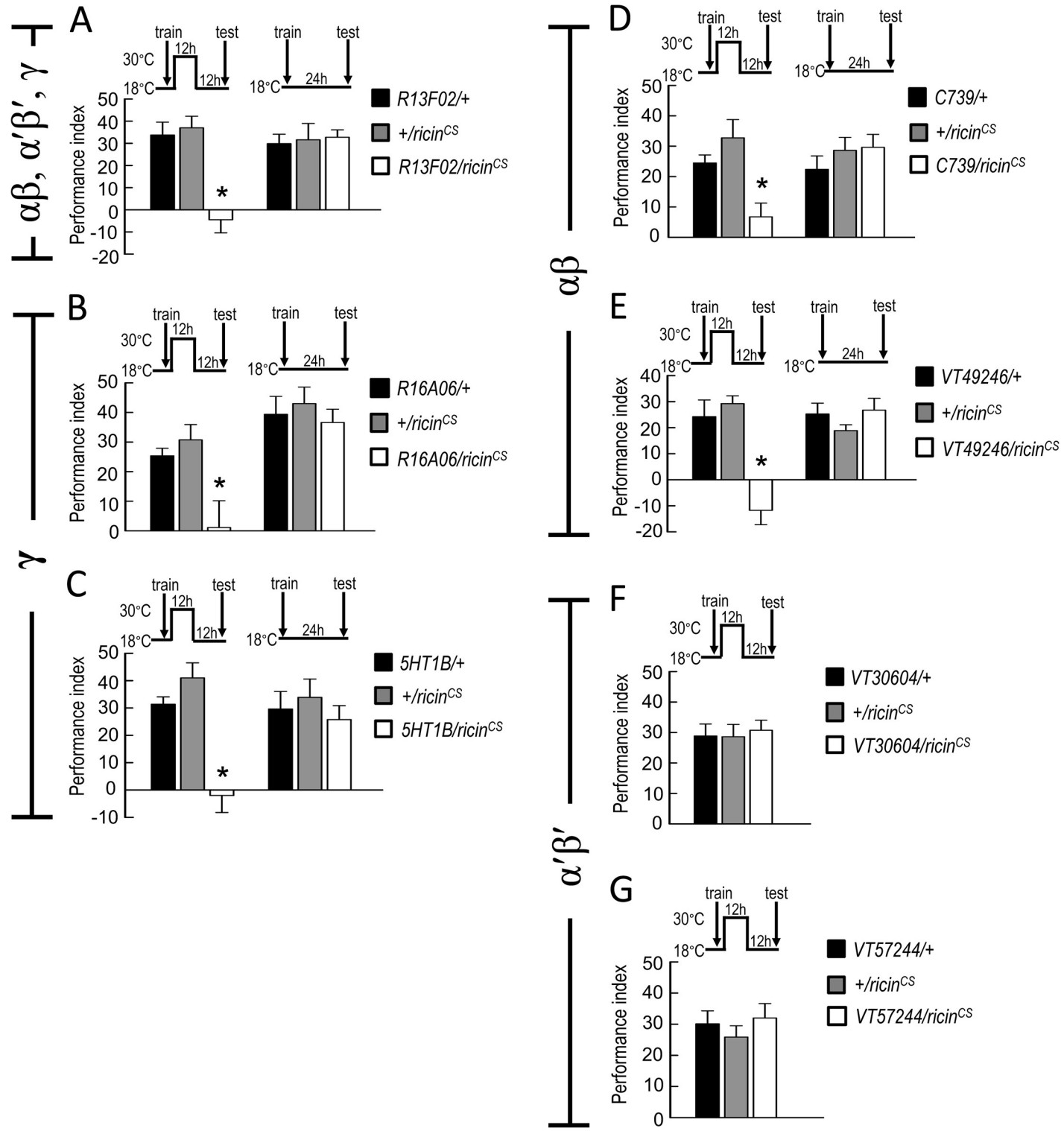

**Fig 1. Blocking *de novo* protein synthesis in αβ and γ neurons during memory formation disrupts wLTM.** (A) Blocking protein synthesis in MB neurons using *R13F02-GAL4* to drive the expression of activated RICIN[CS] during memory formation impaired wLTM. After training at 18˚C, the flies were moved to 30˚C for 12 h to activate RICIN[CS], moved back to 18˚C for another 12 h to inactivate RICIN[CS], and then tested for memory retention at 18˚C (left panel). Each value represents mean ± SEM (N = 8). *, p < 0.05; one-way ANOVA followed by Tukey's test. The 24-hour water-reward memory was normal with inactive RICIN[CS] (18˚C) expression in MB neurons all the way during behavioral assay (right panel). Each value represents mean ± SEM (N = 8). p > 0.05; one-way ANOVA. (B-C) Blocking protein synthesis during memory formation in γ neurons using *R16A06-GAL4* (B) or *5HT1B-GAL4* (C) to drive the expression of activated RICIN[CS] (30˚C, left panel)

impaired wLTM. Each value represents mean ± SEM (N = 8). *, p < 0.05; one-way ANOVA followed by Tukey's test. (D-E) Blocking protein synthesis in αβ neurons during memory formation using *C739-GAL4* (D) or *VT49246-GAL4* (E) to drive the expression of activated RICIN$^{CS}$ (30˚C, left panel) impaired wLTM. Each value represents mean ± SEM (N = 8). *, p < 0.05; one-way ANOVA followed by Tukey's test. (F-G) Blocking protein synthesis in α′β′ neurons using *VT30604-GAL4* (F) or *VT57244-GAL4* (G) to drive the expression of activated RICIN$^{CS}$ (30˚C) did not affect wLTM. Each value represents mean ± SEM (N = 8). p > 0.05; one-way ANOVA.

transgenic flies were trained and kept at 23˚C during the wLTM experiment and were moved to 32˚C only during testing to block the neurotransmitter outputs in the memory retrieval phase. Blocking neurotransmission from αβ surface or γ dorsal neuron subsets during memory retrieval disrupted the wLTM (Figs 3J, 3L, 4D, 4F and S2 Fig), whereas blocking neurotransmission from αβ posterior (Fig 3B and 3D), αβ core (Fig 3F and 3H), or γ main neuron subsets (Fig 4B) during memory retrieval did not affect the wLTM.

## CREB2 activity in αβ surface and γ dorsal neuron subsets is critical for wLTM

Our results showed the CREB2 activity in αβ and γ neuron subsets is essential for wLTM formation (Fig 2). In addition, neurotransmission from αβ surface or γ dorsal neuron subsets is required for wLTM (Figs 3 and 4). These results together led us to investigate whether the CREB2 activity in αβ surface and γ dorsal neurons is essential for wLTM. We, therefore, tested the effect of adult stage-specific *dCREB2b* expression in each αβ and γ neuron subset on wLTM. We used *tub-GAL80$^{ts}$* to limit *UAS-dCREB2b* expression to αβ surface (*VT20803-GAL4*), γ dorsal (*R93G04-GAL4*), αβ posterior (*VT24615-GAL4*), αβ core (*VT8347-GAL4*), or γ main (*R64C08-GAL4*) in adult flies to eliminate the developmental defects of MB neurons.

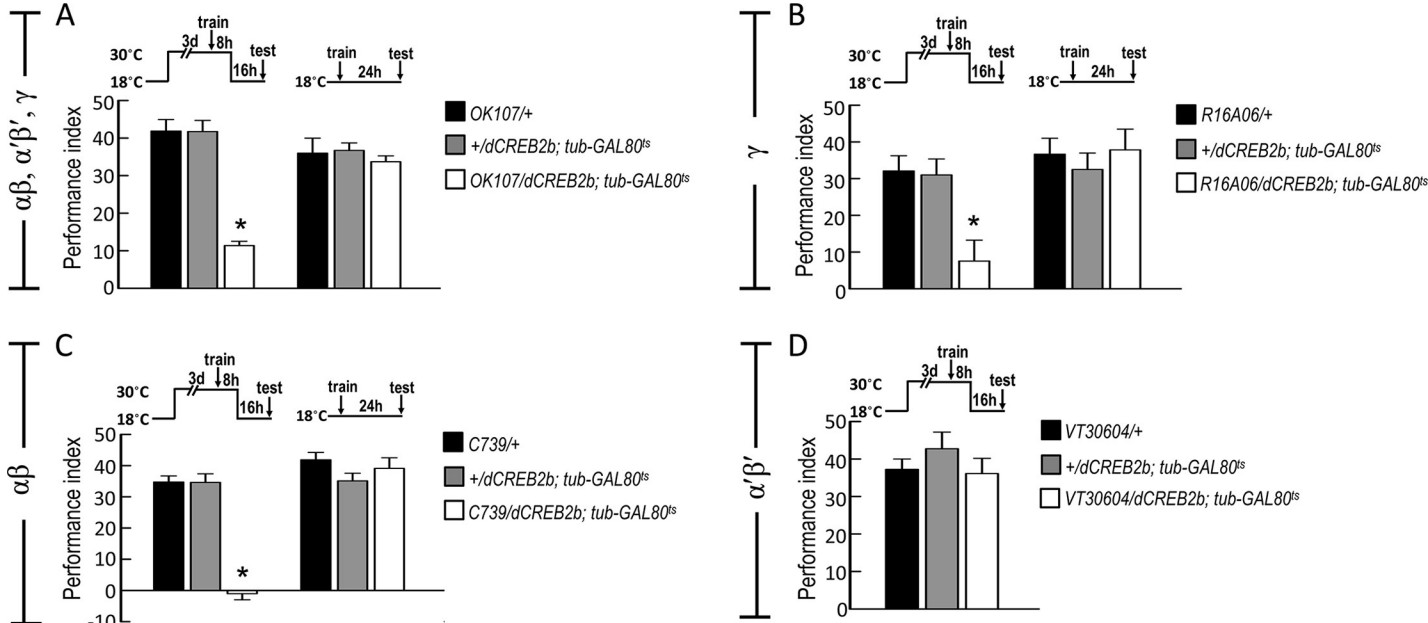

**Fig 2. Adult stage-specific expression of *dCREB2b* in αβ and γ neurons disrupts wLTM.** (A) Inducible expression of *dCREB2b* in MB neurons using *OK107-GAL4* impaired the 24-hour water-reward memory. Adult flies were raised at 18˚C and then transferred to 30˚C for three days before training to remove *tub-GAL80ts* inhibition of *GAL4* activity. Each value represents mean ± SEM (N = 7~9). *, p < 0.05; one-way ANOVA followed by Tukey's test. (B) Inducible expression of *dCREB2b* in γ neurons using *R16A06-GAL4* impaired the 24-hour water-reward memory. Each value represents mean ± SEM (N = 6~11). *, p < 0.05; one-way ANOVA followed by Tukey's test. (C) Inducible expression of *dCREB2b* in αβ neurons using *C739-GAL4* impaired the 24-hour water-reward memory. Each value represents mean ± SEM (N = 8). *, p < 0.05; one-way ANOVA followed by Tukey's test. (D) Inducible expression of *dCREB2b* in α′β′ neurons using *VT30604-GAL4* did not affect the 24-hour water-reward memory. Each value represents mean ± SEM (N = 8). p > 0.05; one-way ANOVA.

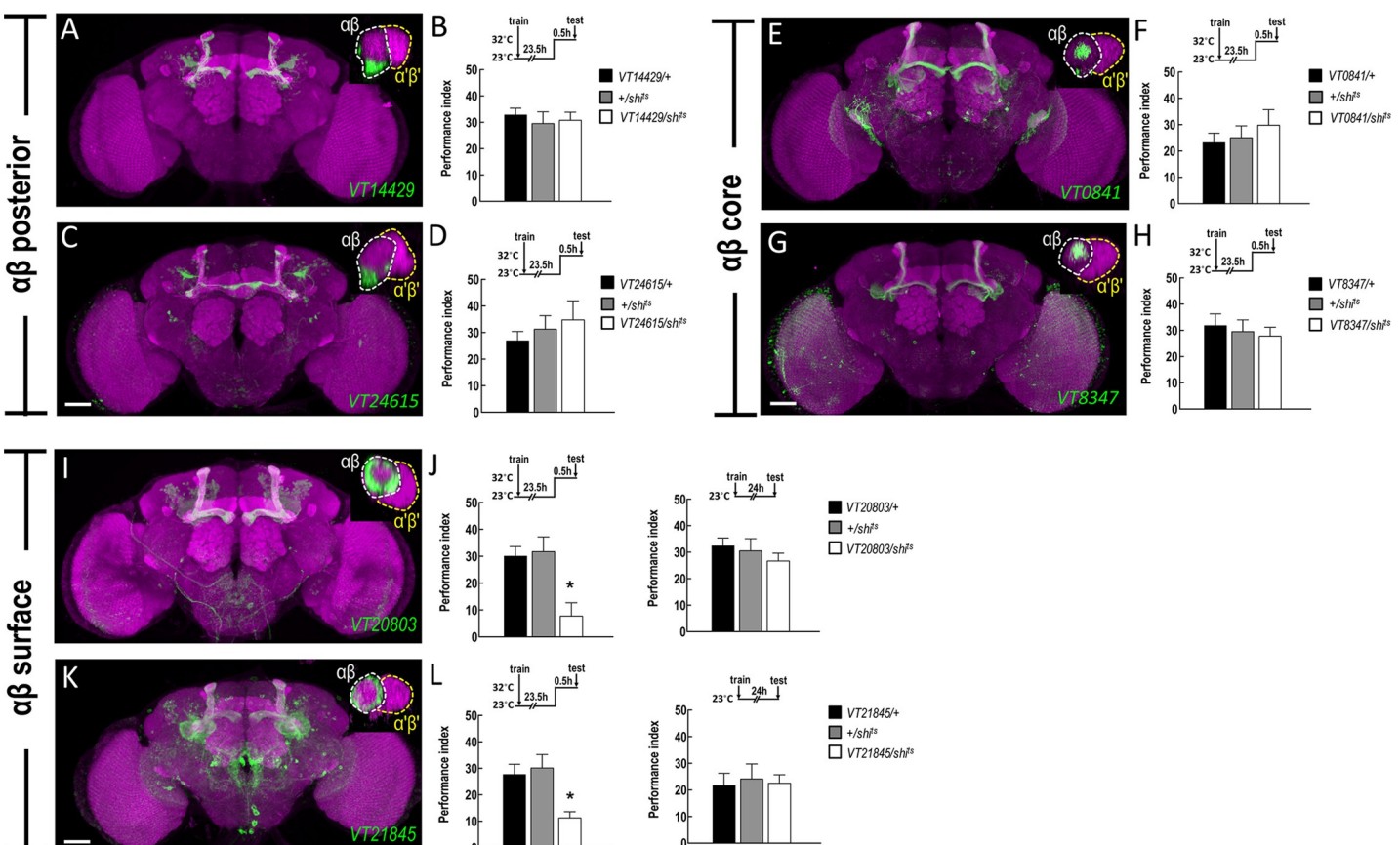

**Fig 3. Retrieval of wLTM requires neurotransmission from αβ surface neurons.** (A) The expression pattern of *VT14429-GAL4* and the high-magnification signal from the horizontal confocal cross section of the MB vertical lobe (top right inset). (B) Blocking neurotransmission from αβ posterior neurons (*VT14429-GAL4*) using *shi^ts* during retrieval did not affect the 24-hour water-reward memory. Each value represents mean ± SEM (N = 8). p > 0.05; one-way ANOVA. (C) The expression pattern of *VT24615-GAL4* and the high-magnification signal from the horizontal confocal cross section of the MB vertical lobe (top right inset). The brain neuropils were immunostained with anti-discs large (DLG) antibody (magenta). Scale bar represents 50 μm. (D) Blocking neurotransmission from αβ posterior neurons (*VT24615-GAL4*) using *shi^ts* during retrieval did not affect the 24-hour water-reward memory. Each value represents mean ± SEM (N = 8). p > 0.05; one-way ANOVA. (E) The expression pattern of *VT0841-GAL4* and the high-magnification signal from the horizontal confocal cross section of the MB vertical lobe (top right inset). (F) Blocking neurotransmission from αβ core neurons (*VT0841-GAL4*) using *shi^ts* during retrieval did not affect the 24-hour water-reward memory. Each value represents mean ± SEM (N = 8). p > 0.05; one-way ANOVA. (G) The expression pattern of *VT8347-GAL4* and the high-magnification signal from the horizontal confocal cross section of the MB vertical lobe (top right inset). The brain neuropils were immunostained with anti-DLG antibody (magenta). Scale bar represents 50 μm. (H) Blocking neurotransmission from αβ core neurons (*VT8347-GAL4*) using *shi^ts* during retrieval did not affect the 24-hour water-reward memory. Each value represents mean ± SEM (N = 8). p > 0.05; one-way ANOVA. (I) The expression pattern of *VT20803-GAL4* and the high-magnification signal from the horizontal confocal cross section of the MB vertical lobe (top right inset). (J) Blocking neurotransmission from αβ surface neurons (*VT20803-GAL4*) using *shi^ts* during retrieval disrupted the 24-hour water-reward memory (left panel). Each value represents mean ± SEM (N = 7). *, p < 0.05; one-way ANOVA followed by Tukey's test. The 24-hour water-reward memory was normal in *VT20803-GAL4 > UAS-shi^ts* flies as compared to their internal control groups in 23°C (right panel). Each value represents mean ± SEM (N = 6). p > 0.05; one-way ANOVA. (K) The expression pattern of *VT21845-GAL4* and the high-magnification signal from the horizontal confocal cross section of the MB vertical lobe (top right inset). The brain neuropils were immunostained with anti-DLG antibody (magenta). Scale bar represents 50 μm. (L) Blocking neurotransmission from αβ surface neurons (*VT21845-GAL4*) using *shi^ts* during retrieval disrupted the 24-hour water-reward memory (left panel). Each value represents mean ± SEM (N = 8). *, p < 0.05; one-way ANOVA followed by Tukey's test. The 24-hour water-reward memory was normal in *VT21845-GAL4 > UAS-shi^ts* flies as compared to their internal control groups in 23°C (right panel). Each value represents mean ± SEM (N = 8). p > 0.05; one-way ANOVA.

We found that adult stage-specific expression of *dCREB2b* in αβ surface or γ dorsal neuron subsets disrupted the wLTM (Fig 5A and 5B). However, adult stage-specific expression of *dCREB2b* in αβ core, αβ posterior, or γ main neuron subsets did not affect wLTM formation (Fig 5C–5E). In addition, we found that blocking *de novo* protein synthesis in αβ surface or γ dorsal neuron subsets also disrupted the wLTM (S3 Fig). Taken together, these results suggest that wLTM formation requires CREB2 activity and *de novo* protein synthesis specifically in αβ surface and γ dorsal neurons.

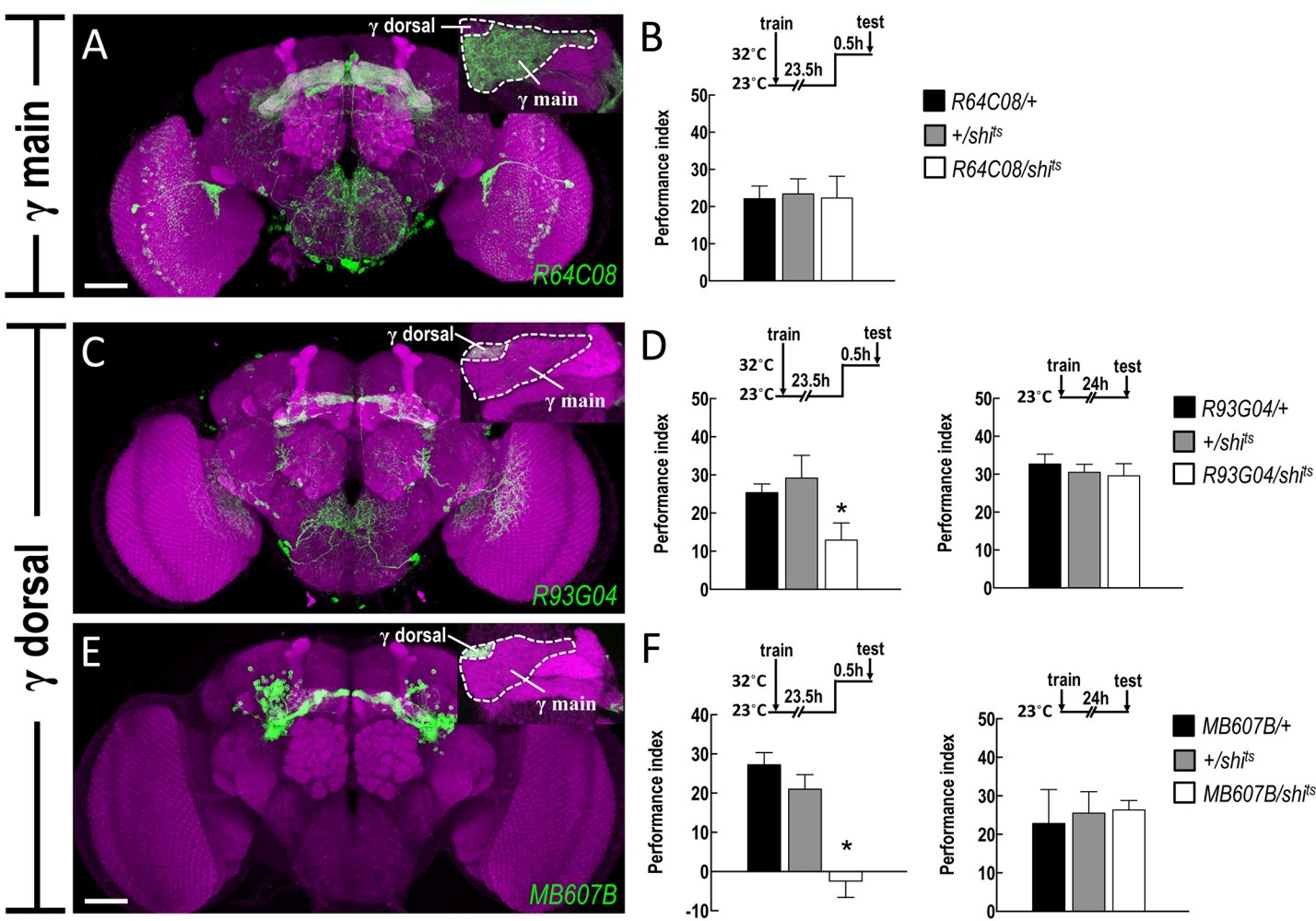

**Fig 4. wLTM retrieval requires neurotransmission from γ dorsal neurons.** (A) The expression pattern of *R64C08-GAL4* and the high-magnification signal from the frontal confocal cross section of the MB horizontal lobe (top right inset). The brain neuropils were immunostained with anti-DLG antibody (magenta). Scale bar represents 50 μm. (B) Blocking neurotransmission from γ main neurons (*R64C08-GAL4*) using *shi*^ts^ during retrieval did not affect wLTM. The 24-hour water-reward memory was normal in *R64C08-GAL4 > UAS-shi*^ts^ flies as compared to their internal control groups in 32°C. Each value represents mean ± SEM (N = 10). p > 0.05; one-way ANOVA. (C) The expression pattern of *R93G04-GAL4* and the high-magnification signal from the frontal confocal cross section of the MB horizontal lobe (top right inset). (D) Blocking neurotransmission from γ dorsal neurons (*R93G04-GAL4*) using *shi*^ts^ during retrieval disrupted wLTM (left panel). Each value represents mean ± SEM (N = 12). *, p < 0.05; one-way ANOVA followed by Tukey's test. The 24-hour water-reward memory was normal in *R93G04-GAL4 > UAS-shi*^ts^ flies as compared to their internal control groups in 23°C (right panel). Each value represents mean ± SEM (N = 6). p > 0.05; one-way ANOVA. (E) The expression pattern of *MB607B-GAL4* and the high-magnification signal from the frontal confocal cross section of the MB horizontal lobe (top right inset). The brain neuropils were immunostained with anti-DLG antibody (magenta). Scale bar represents 50 μm. (F) Blocking neurotransmission from γ dorsal neurons (*MB607B-GAL4*) using *shi*^ts^ during retrieval disrupted wLTM (left panel). Each value represents mean ± SEM (N = 8~9). *, p < 0.05; one-way ANOVA followed by Tukey's test. The 24-hour water-reward memory was normal in *MB607B-GAL4 > UAS-shi*^ts^ flies as compared to their internal control groups in 23°C (right panel). Each value represents mean ± SEM (N = 6). p > 0.05; one-way ANOVA.

## Different dynamics of cellular calcium response to training odor in αβ surface and γ dorsal neurons

LTM formation occurs through a series of changes within neurons, which is essential to encode the relevant sensory inputs. The cellular calcium response to the associative stimuli can be any neural activity change induced by learning, which alters the neuronal response to the sensory information. This change allows the neurons to become more or less excitable, which in turn makes them more or less capable to fire an action potential. Functional *in vivo* calcium imaging has provided insights into neuronal responses to specific odors in a living fly brain by

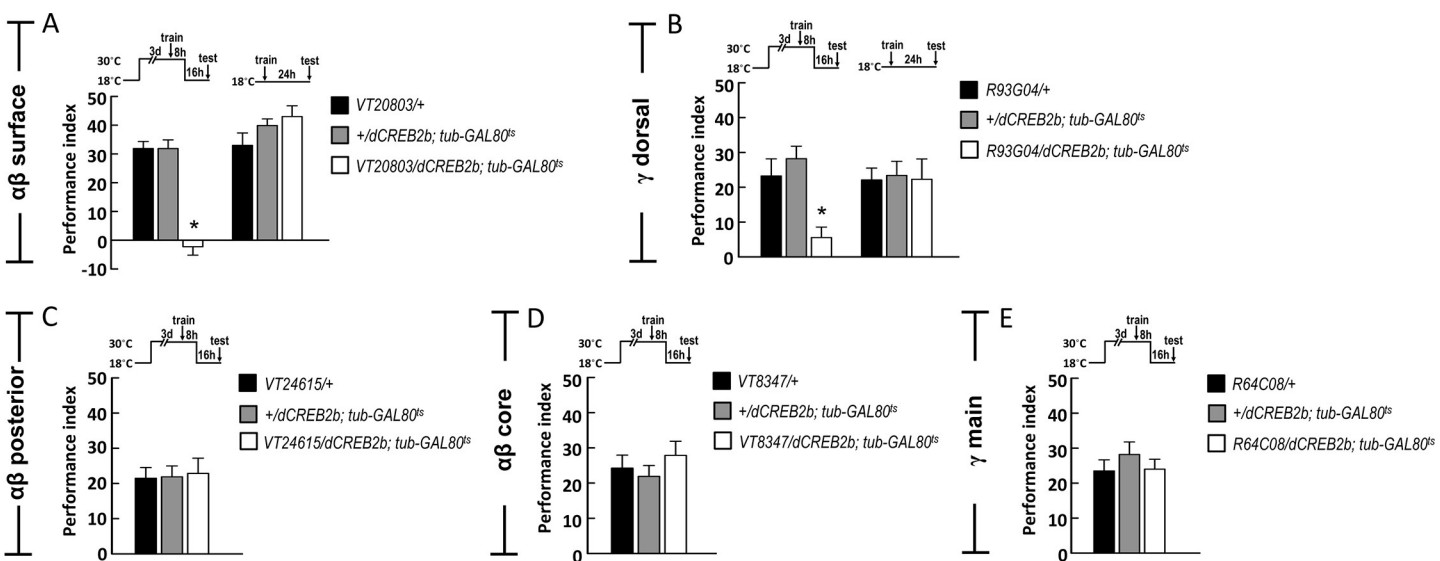

**Fig 5. Adult stage-specific expression of *dCREB2b* in αβ surface or γ dorsal neurons disrupts wLTM.** (A) Inducible expression of *dCERB2b* in αβ surface neurons (*VT20803-GAL4*) disrupted wLTM. The 24-hour water-reward memory was disrupted in flies carrying *VT20803-GAL4 > UAS-dCREB2b; tub-GAL80ᵗˢ* compared to their internal controls. Each value represents mean ± SEM (N = 8). *, p < 0.05; one-way ANOVA followed by Tukey's test. (B) Inducible expression of *dCERB2b* in γ dorsal neurons disrupted wLTM. The 24-hour water-reward memory was disrupted in flies carrying *R93G04-GAL4 > UAS-dCREB2b; tub-GAL80ᵗˢ* compared to their internal controls. Each value represents mean ± SEM (N = 8~10). *, p < 0.05; one-way ANOVA followed by Tukey's test. (C) Inducible expression of *dCERB2b* in αβ posterior (*VT24615-GAL4*) did not affect wLTM. The 24-hour water-reward memory was normal in flies carrying *VT24615-GAL4 > UAS-dCREB2b; tub-GAL80ᵗˢ* compared to their internal controls. Each value represents mean ± SEM (N = 8~10). p > 0.05; one-way ANOVA. (D) Inducible expression of *dCERB2b* in αβ core (*VT8347-GAL4*) neurons did not affect wLTM. The 24-hour water-reward memory was normal in flies carrying *VT8347-GAL4 > UAS-dCREB2b; tub-GAL80ᵗˢ* compared to their internal controls. Each value represents mean ± SEM (N = 8~10). p > 0.05; one-way ANOVA. (E) Inducible expression of *dCERB2b* in γ main (*R64C08-GAL4*) neurons did not affect wLTM. The 24-hour water-reward memory was normal in flies carrying *R64C08-GAL4 > UAS-dCREB2b; tub-GAL80ᵗˢ* compared to their internal controls. Each value represents mean ± SEM (N = 8~10). p > 0.05; one-way ANOVA.

allowing the examination of increased or decreased calcium responses to training odor after conditioning [19]. Several previous studies have shown that different calcium responses to training odor (called memory trace) occurs in the distinct axons of MB neurons at different time windows after odor/shock conditioning [20–23]. We therefore investigated whether the thirsty flies could form a memory trace in MB neurons after odor/water conditioning, and the paired and unpaired training protocols were used for *in vivo* calcium imaging analysis. For the paired training group, the thirsty flies carrying *UAS-GCaMP6m* plus *VT20803-GAL4* received CS− odor without water, followed by exposure to the CS+ odor with water. For the unpaired training group, the thirsty flies received CS− odor without water, followed by exposure to the CS+ odor without water, and the water was provided 1-minute after CS+ odor delivery. A significantly increased calcium response to training odor was observed in the α-lobe region of the αβ surface neurons, but not in the αβ core neurons at 24-hour after odor/water association in thirsty flies (Fig 6 and S4 Fig). The increased cellular calcium response to training odor was eliminated in the same flies carrying *UAS-dCREB2b* transgene, suggesting that this 24-hour memory trace is CREB2-dependent (Fig 6). Moreover, the increased cellular calcium response to training odor occurred not only in the α-lobe but also in the β-lobe of surface neurons (S5 Fig).

In addition, thirsty flies carrying *UAS-GCaMP6m* plus *R93G04-GAL4* were trained to associate an odor with water, and 24-hour odor responses were recorded. Intriguingly, we found that naïve flies showed decreased calcium responses to odor in the γ dorsal neurons (Fig 7 and S6 Fig). However, in the paired training group, the decrease in calcium response to training odor was even stronger compared to the naïve flies 24-hour after training (Fig 7). This

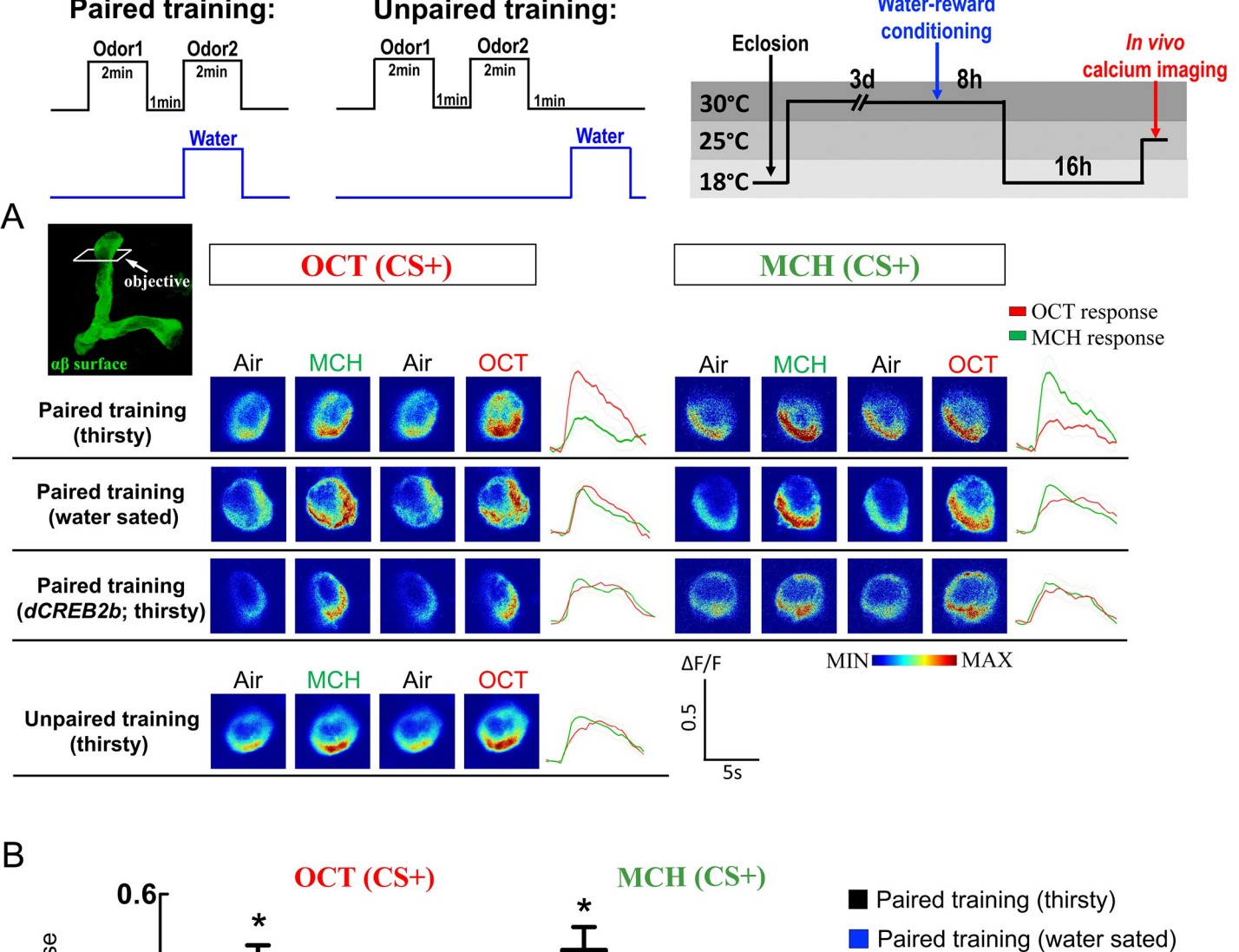

**Fig 6. αβ surface neurons show a CREB2-dependent increased cellular calcium response to training odor at 24-hour post-conditioning.** Top diagram illustrating the protocols of paired training, unpaired training, and *in vivo* calcium imaging. (A) The GCaMP6 response 24-hour after water-reward conditioning was assayed in αβ surface neurons (the image-recording region is showed in the top left figure). For the paired training group: flies received CS− odor without water-reward (US), followed by exposure to the CS+ odor with water-reward. For the unpaired training group: flies received CS− odor without water-reward, followed by exposure to CS+ odor without water-reward, and water-reward was delivered 1-minute after CS+ odor. Odor/water paired training induced additional increment of GCaMP6 responses in the α-lobe region of αβ surface neurons to the training odor [OCT-trained flies: OCT (CS+), MCH-trained flies: MCH (CS+)] in thirsty-state. This additional increased calcium response was abolished in water-sated state (flies allowed to drink water for 30 minutes before *in vivo* calcium imaging recording) or in thirsty flies with *dCREB2b* expression in αβ surface neurons. (B) Quantification of the GCaMP6 responses to the training odor (CS+) relative to the non-training odor (CS−) in the α-lobe region of αβ surface neurons in OCT-trained (left panel) or MCH-trained (right panel) flies at 24-hour post-conditioning. The Log ratios of the CS+ response to the CS− response were calculated using the peak response amplitudes. Each value represents mean ± SEM (N = 8~12). *, p < 0.05; significantly different from zero; one sample t-test. Genotype: (1) Flies carrying *UAS-GCaMP6m/tub-GAL80ts;VT20803-GAL4/+* transgenes were used for Paired training (thirsty), Paired training (water

sated), and Unpaired training (thirsty) experiments, (2) Flies carrying *UAS-GCaMP6m/tub-GAL80^ts^; VT20803-GAL4/UAS-dCREB2b* transgenes were used for Paired training (*dCREB2b*; thirsty) experiment.

training-induced additional decrease in calcium response to training odor was eliminated in the same flies carrying *UAS-dCREB2b* transgene. These results suggest that thirsty flies show an increased cellular calcium response to training odor in αβ surface neurons and a decreased cellular calcium response to training odor in γ dorsal neurons at 24-hour after odor/water conditioning, which implies that different neuronal dynamics of wLTM trace occur in αβ surface and γ dorsal neurons (Fig 8).

## Discussion

CREB-dependent gene transcription is critical for memory formation, especially LTM, in both vertebrates and invertebrates [4, 24]. Several previous studies in *Drosophila* suggest that LTM formation requires CREB-dependent gene transcription followed by *de novo* protein synthesis [1, 3, 6, 9]. Moreover, it has been reported that CREB2 activity in both αβ and α′β′ neurons is critical for appetitive LTM produced by sugar-reward conditioning [9, 25]. Here, we observed that wLTM formation requires *de novo* protein synthesis in the αβ and γ neurons (Fig 1). Moreover, adult stage-specific expression of the CREB2 repressor (*dCREB2b*) in αβ or γ neurons disrupts wLTM, whereas CREB2 activity in α′β′ neurons is dispensable (Fig 2). Food or water deprivation is necessary to induce the motivational drive in flies to form sugar- or water-reward LTM since different motivational drives are critical for distinct memories. It has been shown that individual internal motivational inputs for sugar and water are delivered via distinct MB input neurons, which finally induce sugar- or water-reward LTM in different MB neuron subsets [1, 9, 26, 27]. Previous studies together with our current findings suggest that CREB2 activity is required for both sugar- and water-reward LTMs, however, these LTMs are processed in different MB circuits [9, 25].

Neurotransmission from αβ neurons is required for both shock-punitive and sugar-reward LTMs retrieval [28], whereas neurotransmission from γ neurons is dispensable for retrieval of both shock-punitive and sugar-reward LTMs [28, 29]. However, our previous study showed that the neurotransmission from αβ and γ neurons is required for wLTM retrieval [1] suggesting that wLTM is different from the other types of olfactory associative LTMs at MB circuit levels. The αβ and γ neurons are classified into αβ core, αβ surface, αβ posterior, γ main, and γ dorsal neuron subsets according to the morphology of their axons [15]. It has been shown that neurotransmission from the combination of MB αβ surface and αβ posterior subsets is necessary for the retrieval of both shock-punitive and sugar-rewarded LTMs [30]. Another study suggests that the neurotransmission from αβ surface and αβ core subsets is required for the retrieval of sugar-reward LTM [31]. These results imply that several different subsets of αβ neurons participate in the retrieval of *Drosophila* sugar-reward LTM. Here, we showed that neurotransmission only from αβ surface is necessary for wLTM retrieval, whereas the αβ core and αβ posterior subdivisions are dispensable (Fig 3 and S2 Fig). Contrary to the sugar reward conditioning in which γ neurons are dispensable for the retrieval of sugar-reward LTM [28, 29], wLTM retrieval requires neurotransmission from γ dorsal but not from γ main neuron subset (Fig 4 and S2 Fig). Taken together, these results imply that γ dorsal neuronal activity is specifically required for wLTM retrieval but not for sugar-reward LTM. Adult stage-specific expression of *dCREB2b* or blocking *de novo* protein synthesis in αβ surface and γ dorsal neurons disrupts wLTM, further suggesting the crucial role of αβ surface and γ dorsal neurons in *Drosophila* wLTM process (Fig 5 and S3 Fig).

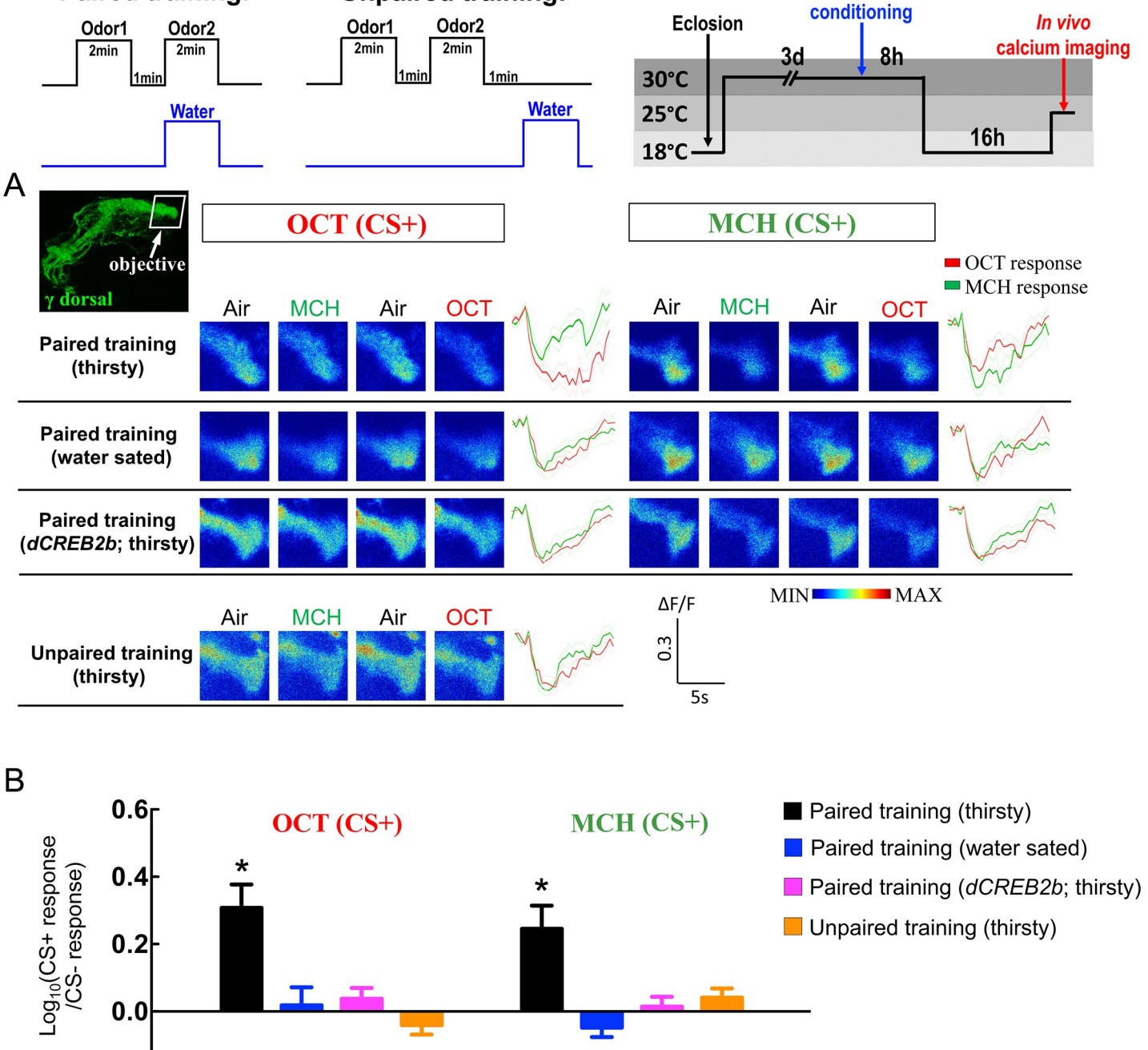

**Fig 7. γ dorsal neurons show a CREB2-dependent decreased cellular calcium response to training odor at 24-hour post-conditioning.** Top diagram illustrating the protocols of paired training, unpaired training, and *in vivo* calcium imaging. (A) The GCaMP6 response 24-hour after water-reward conditioning was assayed in γ dorsal neurons (the image-recording region is showed in the top left figure). For the paired training group: flies received CS− odor without water-reward (US), followed by exposure to the CS+ odor with water-reward. For the unpaired training group: flies received CS− odor without water-reward, followed by exposure to CS+ odor without water-reward, and the water-reward was delivered 1-minute after CS+ odor. Odor/water paired training induced an additional decrease in GCaMP6 responses in the γ-lobe region of the γ dorsal neurons to the training odor [OCT-trained flies: OCT (CS+), MCH-trained flies: MCH (CS+)] in thirsty-state. This additional decreased calcium response was abolished in water-sated state (flies allowed to drink water for 30 minutes before *in vivo* calcium imaging recording) or in thirsty flies with *dCREB2b* expression in γ dorsal neurons. (B) Quantification of the GCaMP6 responses to the training odor (CS+) relative to the non-training odor (CS−) in the γ-lobe region of γ dorsal neurons in OCT-trained (left panel) or MCH-trained (right panel) flies at 24-hour post-conditioning. The Log ratios of the CS+ response to the CS− response were calculated using the peak response amplitudes. Each value represents mean ± SEM (N = 9~11). *, p < 0.05; statistically significantly different from zero; one sample t-test. Genotype: (1) Flies carrying *UAS-GCaMP6m/tub-GAL80^{ts}; R93G04-GAL4/+* transgenes were used for Paired training (thirsty), Paired training

(water sated), and Unpaired training (thirsty) experiments, (2) Flies carrying *UAS-GCaMP6m/tub-GAL80^{ts}; R93G04-GAL4/UAS-dCREB2b* transgenes were used for Paired training (*dCREB2b*; thirsty) experiment.

Our previous study suggests that PAM-β′1 neurons convey the water-rewarding event as the US signal to the MB β′ lobes, and the neurotransmission in α′β′ neurons is required for wLTM consolidation [1]. How the wLTM is transferred from α′β′ neurons and finally stored in αβ surface and γ dorsal neurons through system consolidation is still unclear. In both shock-punitive and sugar-reward LTMs, the neurotransmission in α′β′, γ, and, αβ neurons is required for at least 3 hours after conditioning, but the expression of 24-hour shock-punitive or sugar-rewarded memories only requires neurotransmission in αβ neurons [28]. In addition, the expression of DopR1, a D1-like dopamine receptor, in the γ neurons is sufficient to fully support the shock-punitive STM and LTM in DopR1 mutant background (*dumb^2*), suggesting

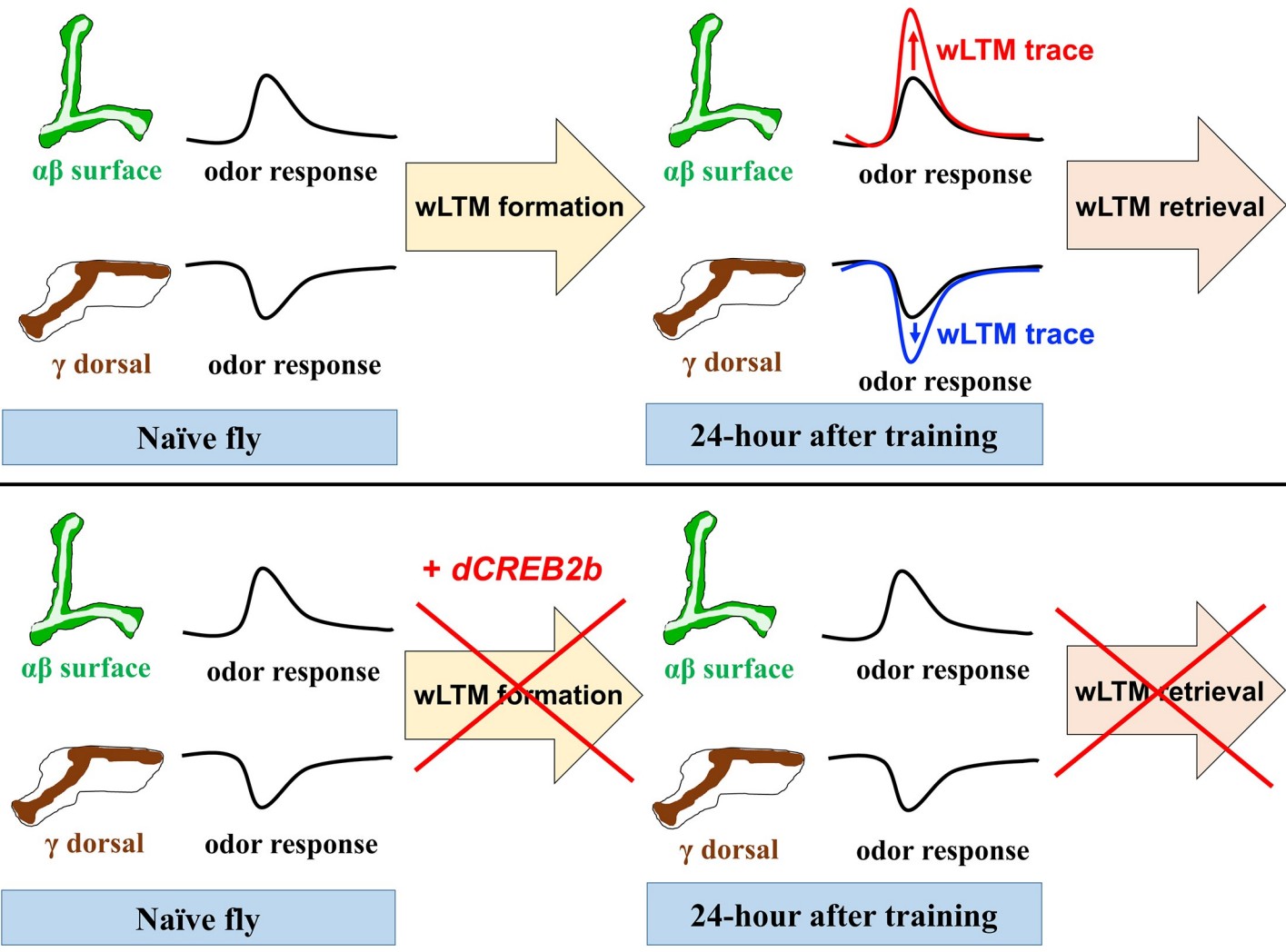

**Fig 8. A brain circuitry model of *CREB2* and wLTM.** The αβ surface neurons show an evoked calcium response to odors while the γ dorsal neurons show a decreased calcium response to odors in naïve flies. After odor/water association which induces the formation of wLTM, calcium response to training odor was further increased in αβ surface neurons, whereas the calcium response was further decreased in the γ dorsal neurons 24-hour after training. These increased and decreased cellular calcium responses to training odor in αβ surface and γ dorsal neurons, respectively, were abolished by *dCREB2b* expression, suggesting that training-induced cellular calcium responsive changes in distinct MB subsets form wLTM traces with different neuronal dynamics in thirsty fly.

that dopamine-mediated odor/shock association is registered in γ neurons and finally stabilized and maintained in αβ neurons for long-term storage [32]. It might be possible that other MB input neurons (i.e., PAM) convey water rewarding events to MB αβ surface and γ dorsal neurons individually during water drinking in thirsty flies, which allow the initial odor/water associations in αβ surface and γ dorsal neurons [1, 11]. The synaptic output from α′β′ neurons is required for stabilizing and strengthening the associative plasticity in αβ surface and γ dorsal neurons during wLTM consolidation. How the α′β′ neurons communicate with αβ surface and γ dorsal neurons in wLTM consolidation phase, remains unclear. The intrinsic MB neurons, also known as the dorsal paired medial (DPM), arborize widely throughout the MB lobes and play a critical role in modulating MB functions [33–38], including the consolidation of olfactory memories [9, 39, 40]. It might also be possible that DPM neurons receive the input from α′β′ neurons and transmit to αβ surface and γ dorsal neurons during wLTM consolidation. In addition, the output of MB Kenyon cells is conveyed to 34 neurons (MB output neuron, MBON) of 21 cell types per brain hemisphere, and 9 MBONs receive the inputs from α′β′ neurons [41]. Another possibility is that the activity from α′β′ neurons is transmitted to αβ surface and γ dorsal neurons via the relevant α′β′ MBONs and their downstream neurons. Therefore, it is noteworthy to test the physiological roles of DPM neurons and α′β′ MBONs during consolidation phase of wLTM.

A significant increase in cellular calcium response to training odor in the αβ surface neurons, but not in other αβ neuron subsets, was observed 24-hour after water-reward conditioning (Fig 6, S4 Fig and S5 Fig). These results are consistent with our behavioral study showing that neurotransmission from αβ surface neurons is required for wLTM retrieval, whereas the αβ core and αβ posterior neuron subsets are dispensable (Fig 3). This training-induced increased calcium response was abolished in water-sated or *dCREB2b* expressing flies (Fig 6). A previous study showed that the fly forms α-lobe branch-specific aversive LTM trace 24-hour after odor/shock conditioning [20]. In our recent study, we also observed the α-lobe branch-specific aversive anesthesia-resistant memory (ARM) trace 3-hour after odor/shock conditioning [23]. Intriguingly, we found that both α- and β- lobes of the surface neurons show increased calcium response to training odor 24-hour after odor/water conditioning, suggesting that the wLTM trace is not specific to the α-lobe branch (Fig 6 and S5 Fig).

An increased GCaMP response to training odor in MB γ neurons is observed at 24-hour after ten sessions of spaced odor/shock training, and this increased calcium response is abolished by expressing *dCREB2b* in γ neurons throughout the fly development [21]. Here, we found that γ main neuron subset shows an evoked calcium response to both odors, but no further increased calcium response to the training odor at 24-hour after water-reward conditioning was observed (S7 Fig). A previous study suggests that γ dorsal neurons respond to visual stimuli, which is required for visual, but not for aversive olfactory memory in *Drosophila* [42]. However, we observed a decreased calcium response to odor stimuli in the γ dorsal lobe (Fig 7 and S6 Fig), which is consistent with the electrophysiological study showing slow inhibitory responses to odor stimuli in the γ dorsal neurons [42]. Interestingly, we observed a further decrease in calcium responses to the training odor in the γ dorsal neurons 24-hour after water-reward conditioning (Fig 7). This training-induced additional decrease in the calcium response is abolished in the water-sated or acutely *dCREB2b* expressing flies, suggesting a type of wLTM trace in the γ dorsal neurons different from the αβ surface neurons (Figs 6, 7 and 8). Since blocking neurotransmission from the γ dorsal neuron subset during memory retrieval disrupts wLTM, why the γ dorsal neurons show additionally suppressed calcium response to training odor, needs to be answered. One possible explanation is that odor/water association alters the olfactory response of MB neurons to the training odor, and this change can be represented by an increased or decreased calcium response as compared to the response of the non-

training odor (memory traces). The training-induced differences in odor responsive levels in the MB allow the flies to distinguish two odors by increasing the contrast and perform appropriate behavioral output during testing. However, *shi^ts* abolishes the increased or decreased training-odor responses thereby eliminating the contrast between odors, and consequently, the flies could not distinguish two odors and make appropriate behavioral output during testing.

In conclusion, our study show that αβ surface and γ dorsal neuron subsets regulate *Drosophila* wLTM. Blocking neurotransmission from αβ surface or γ dorsal neurons only abolishes wLTM retrieval but does not affect the olfactory acuity or water preference in thirsty flies (Figs 3, 4 and S2 Fig). Further, adult stage-specific expression of *dCREB2b* or blocking *de novo* protein synthesis in αβ surface and γ dorsal neurons disrupts wLTM (Fig 5 and S3 Fig). Different dynamics of cellular wLTM traces are formed in the αβ surface and γ dorsal neurons, which are blocked by *dCREB2b* expression (Figs 6, 7 and 8). Taken together, our results reveal a small population of MB neurons that encode wLTM in the brain and provide a broader view of the olfactory memory process in fruit flies.

## Materials and methods

### Fly stocks

All the flies were reared on standard cornmeal food at 25°C and 60% relative humidity on a 12 h:12 h, light-dark cycle. *R13F02-GAL4, R16A06-GAL4, 5HT1B-GAL4, C739-GAL4, VT44966-GAL4, VT30604-GAL4, VT57244-GAL4, tub-GAL80^ts; UAS-dCREB2b, UAS-mCD8::GFP; UAS-mCD8::GFP, UAS-shi^ts*, and *UAS-GCaMP6m* fly strains have been used as described previously [1, 23, 35, 37, 43–45]. *UAS-ricin^CS* and *MB607B-GAL4* flies were obtained from Ann-Shyn Chiang. *OK107-GAL4, VT14429-GAL4, VT24615-GAL4, VT20803-GAL4, VT21845-GAL4, VT0841-GAL4, VT8347-GAL4, R64C08-GAL4*, and *R93G04-GAL4* fly strains were obtained from the Vienna *Drosophila* Resource Center and Bloomington *Drosophila* Stock Center.

### Immunohistochemistry and brain imaging

Fly brains were dissected in isotonic phosphate buffered saline (PBS) and transferred to 4% paraformaldehyde (PFA) for fixation for 20 min at 25°C. The samples were then incubated in penetration and blocking buffer (PBS containing 2% Triton X-100 and 10% normal goat serum) for 2 h at 25°C. The brain samples were also subjected to a degassing procedure during the 2 h penetration and blocking period. Thereafter, the brains were incubated in 1:10 diluted mouse 4F3 anti-discs large (DLG) monoclonal antibody (AB 528203, Developmental Studies Hybridoma Bank, University of Iowa) at 25°C for 24 h. After washing in PBS-T (PBS containing 1% Triton X-100), the samples were incubated in 1:200 biotinylated goat anti-mouse IgG (31800, Thermo Fisher Scientific) at 25°C for 24 h. Next, the brain samples were washed and incubated in 1:500 Alexa Fluor 635 streptavidin (S32364, Thermo Fisher Scientific) at 25°C for 24 h. After intensive washing in PBS-T, the brains were cleared and mounted in FocusClear (FC-101, CelExplorer) for imaging. The brains were imaged under a Zeiss LSM 700 confocal microscope with either a 40× C-Apochromat water-immersion objective (N.A. value, 1.2; working distance, 220 μm) or a 63× glycerin-immersion objective (N.A. value, 1.4; working distance, 170 μm). The confocal pinhole (optical section) was set at 2 μm for images taken with the 40× objective lens and at 1.5 μm when imaging at 63× objective lens. Some slides were imaged twice to overcome the limited field of view; one image for each brain hemisphere, with an overlap in between. Two parallel brain image stacks were combined into a single dataset using the overlapping region to align the two confocal stacks by ZEN software.

## Behavioral assay

The wLTM assay has been described in our previous study [1]. Briefly, the flies were water deprived by keeping them in a glass milk bottle containing a 6 cm × 3 cm piece of dry sucrose-soaked filter paper for 16 h before water-reward conditioning assay. The flies were first exposed to one odor for 2 min (CS−: 4-methylcyclohexanol (MCH) or 3-octanol (OCT)) in a tube lined with dry filter paper, followed by 1 min of fresh room air. The flies were then transferred to another tube that contained a water-soaked filter paper and exposed to a second odor for another 2 min (CS+: OCT or MCH). After that, the flies were transferred to a clean training tube and exposed to fresh room air for 1 min. The trained flies were kept in a plastic vial that contained a 1.5 cm × 3 cm piece of a dried sucrose-soaked filter paper during the 24 h interval. During the testing phase, the flies were presented with a choice between CS+ and CS− odors in a T-maze for 2 min. From this distribution, a performance index (PI) was calculated as the number of flies running toward the CS+ odor minus the number of flies running towards the CS− odor, divided by the total number of flies and multiplied by 100. For the calculation of individual PI, naive flies were first trained by pairing water with OCT (CS+), and the $PI_O$ index was calculated. Next, another group of naive flies was trained by pairing water with MCH (CS+), and the $PI_M$ index was calculated. A single PI was calculated from the average of single $PI_O$ and $PI_M$ values. For the $RICIN^{CS}$-related experiments, the flies were kept at 18˚C until eclosion and during the training. The trained flies were then moved to 30˚C for 12 h right after the training, and moved back to 18˚C for another 12 h and then tested. For the control groups, the flies were kept at 18˚C throughout the whole experiment. In $shi^{ts}$-related experiments for blocking neurotransmitter output during wLTM retrieval, flies were trained at 23˚C and then maintained at 23˚C for 23.5 h followed by 32˚C for 0.5 h and tested. For adult stage-specific *dCREB2b* expression with *tub-GAL80^{ts}*, flies were kept at 18˚C until eclosion and then moved to 30˚C for 3 days before water conditioning. The water-reward conditioning was performed at 30˚C and the trained flies were kept at 30˚C for 8 h and then moved back to 18˚C for 16 h and tested. For the control groups, flies were kept at 18˚C throughout the whole experiment.

## Olfactory acuity assay

Groups of approximately 50 water-deprived naïve flies were subjected to a 2-min test trial in the T-maze at a restrictive temperature (32˚C) for olfactory acuity assay. Flies were given a choice between OCT/MCH and 'fresh' room air. The odor avoidance index was calculated as the number of flies in the fresh room-air tube minus the number of flies in the OCT or MCH odor tube, divided by the total number of flies, and multiplied by 100.

## Water preference assay

Groups of approximately 50 water-deprived naïve flies were given 2 min to choose between tubes in a T-maze at a restrictive temperature (32˚C) for water preference assay, with one tube containing a dry filter paper and the other tube containing a water-soaked filter paper. The water preference index was calculated as the number of flies in the tube containing a water-soaked filter paper minus the number of flies in the tube containing a dry filter paper, divided by the total number of flies, and multiplied by 100.

## *In vivo* calcium imaging

The flies carrying (1) *UAS-GCaMP6m/tub-GAL80^{ts}; VT20803-GAL4/+*, (2) *UAS-GCaMP6m/tub-GAL80^{ts}; VT20803-GAL4/UAS-dCREB2b*, (3) *UAS-GCaMP6m/tub-GAL80^{ts};*

*R93G04-GAL4/+*, and (4) *UAS-GCaMP6m/tub-GAL80$^{ts}$; R93G04-GAL4/UAS-dCREB2b* transgenes were kept at 18˚C until eclosion. After, the flies were kept at 30˚C for 3 days before training. The water-reward conditioning was performed at 30˚C and the trained flies were kept at 30˚C for 8 h then moved back to 18˚C for another 16 h. The *in vivo* calcium imaging was performed at 25˚C. The flies carrying (1) *UAS-GCaMP6m/+; VT0841-GAL4/+*, (2) *UAS-G-CaMP6m/+; VT20803-GAL4/+*, (3) *UAS-GCaMP6m/+; R93G04-GAL4/+*, and (4) *UAS-GCaMP6m/+; R64C08-GAL4/+* transgenes were kept at 25˚C throughout the experiment. For the paired training group: flies received CS− odor without water, followed by exposure to the CS+ odor with water. For the unpaired training group: flies received CS− odor without water, followed by exposure to the CS+ odor without water, and water was provided 1 min after the CS+ odor delivery.

The flies from the paired or unpaired training groups were immobilized in a 250-ml pipette tip. A window was opened on the head capsule using fine tweezers and a drop of adult hemolymph-like (AHL) saline (108 mM NaCl, 5 mM KCl, 2 mM CaCl$_2$, 8.2 mM MgCl$_2$, 4 mM NaHCO$_3$, 1 mM NaH$_2$PO$_4$, 5 mM trehalose, 10 mM sucrose, and 5 mM HEPES (pH 7.5, 265 mOsm)) was added immediately to prevent dehydration. After removing the small trachea and excessive fat with fine tweezers, the fly and the pipette tip were fixed to a coverslip by tape, and a 40× water immersion objective (W Plan-Apochromat 40×/1.0 DIC M27) was used for imaging. Odorants were delivered via a custom-made odor delivery device. The frames were aligned using a lightweight SIFT-implementation to correct the motion artifacts. The calcium responses were calculated as the mean change in fluorescence intensity (ΔF/F) in the 0.1–5 s window after stimulus onset. Time-lapse recordings of changes in GCaMP intensity before and after odor delivery were performed under a Zeiss LSM700 microscope with an excitation laser (488 nm) and a detector for emissions passing through a 555 nm short-pass filter. An optical slice with a resolution of 512 × 512 pixels was continuously monitored for 45 s at a rate of 2 frames per second. Regions of interest were manually assigned to anatomically different regions of the MB lobes. To evaluate responses to different odors in flies, we calculated the changes in GCaMP6 fluorescence as $\Delta F(Ft–F_0)/F(F_0)$. Changes in GCaMP6 fluorescent intensity for the CS+ vs. CS− odors were calculated as $\log_{10}(\Delta F_{CS+}/\Delta F_{CS−})$. The intensity maps were generated using ImageJ software for all functional calcium imaging studies.

## Statistical analysis

All the raw data were analyzed parametrically using the Prism 7.0 software (GraphPad). The raw data from more than two groups were evaluated by one-way analysis of variance (ANOVA) and Tukey's multiple comparison test. The raw data from only two groups were evaluated by paired t-test. The one sample t-test was used to evaluate whether the raw data from one group is significantly different from zero or not. A statistically significant difference was defined as $P < 0.05$. All data were presented as mean ± standard error of the mean (SEM).

## Supporting information

**S1 Fig. GFP expression pattern driven by MB-GAL4s used in this study.** (A) The expression pattern of *R13F02-GAL4*-driven GFP expression in γ, αβ, and α′β′ neurons. (B) The expression pattern of *R16A06-GAL4*-driven GFP expression in γ neurons. (C) The expression pattern of *5HT1B-GAL4*-driven GFP expression in γ neurons. (D) The expression pattern of *C739-GAL4*-driven GFP expression in αβ neurons. (E) The expression pattern of *VT49246-GAL4*-driven GFP expression in αβ neurons. (F) The expression pattern of *VT30604-GAL4*-driven GFP expression in α′β′ neurons. (G) The expression pattern of *VT57244-GAL4*-driven GFP expression in α′β′ neurons. (H) The expression pattern of *OK107-GAL4*-driven GFP

expression in γ, αβ, and α′β′ neurons. The brain neuropils were immunostained with anti-DLG antibody (magenta). Scale bar represents 50 μm.
(TIF)

**S2 Fig. Behavioral control experiments in MB-GAL4s combined with *UAS-shi^{ts}*.** (A) Normal olfactory acuity to OCT or MCH and normal water preference at restrictive temperature (32˚C) in thirsty *VT20803-GAL4 > UAS-shi^{ts}* flies. Each value represents mean ± SEM, (N = 14 for olfactory acuity; N = 8 for water preference). p > 0.05; one-way ANOVA. (B) Normal olfactory acuity to OCT or MCH and normal water preference at restrictive temperature (32˚C) in thirsty *VT21845-GAL4 > UAS-shi^{ts}* flies. Each value represents mean ± SEM, (N = 6 for olfactory acuity; N = 6 for water preference). p > 0.05; one-way ANOVA. (C) Normal olfactory acuity to OCT or MCH and normal water preference at restrictive temperature (32˚C) in thirsty *R93G04-GAL4 > UAS-shi^{ts}* flies. Each value represents mean ± SEM, (N = 6 for olfactory acuity; N = 6 for water preference). p > 0.05; one-way ANOVA. (D) Normal olfactory acuity to OCT or MCH and normal water preference at restrictive temperature (32˚C) in thirsty *MB607B-GAL4 > UAS-shi^{ts}* flies. Each value represents mean ± SEM, (N = 6~8 for olfactory acuity; N = 8~9 for water preference). p > 0.05; one-way ANOVA.
(TIF)

**S3 Fig. Blocking *de novo* protein synthesis in αβ surface and γ dorsal neurons during memory formation disrupts wLTM.** (A) Blocking protein synthesis in αβ surface neurons using *VT20803-GAL4* to drive the expression of activated RICIN^{CS} (30˚C) during memory formation impaired wLTM (left panel). Each value represents mean ± SEM (N = 11). *, p < 0.05; one-way ANOVA followed by Tukey's test. The 24-hour water-reward memory was normal with inactive RICIN^{CS} (18˚C) expression in αβ surface neurons all the way during behavioral assay (right panel). Each value represents mean ± SEM (N = 8~9). p > 0.05; one-way ANOVA. (B) Blocking protein synthesis in γ dorsal neurons using *R93G04-GAL4* to drive the expression of activated RICIN^{CS} (30˚C) during memory formation impaired wLTM (left panel). Each value represents mean ± SEM (N = 11). *, p < 0.05; one-way ANOVA followed by Tukey's test. The 24-hour water-reward memory was normal with inactive RICIN^{CS} (18˚C) expression in γ dorsal neurons all the way during behavioral assay (right panel). Each value represents mean ± SEM (N = 9~10). p > 0.05; one-way ANOVA.
(TIF)

**S4 Fig. αβ core neurons do not form wLTM trace.** (A) The GCaMP6 response 24-hour after water-reward conditioning was assayed in αβ core neurons (the image-recording region is showed in the top left figure). For the paired training group: flies received CS− odor without water-reward (US), followed by exposure to the CS+ odor with water-reward. For the unpaired training group: flies received CS− odor without water-reward, followed by exposure to CS + odor without water-reward, and the water-reward was delivered 1-minute later after CS + odor. Odor/water paired training did not induce wLTM trace 24-hour post-conditioning in the α-lobe region of the αβ core neurons to the training odor [OCT-trained flies: OCT (CS+), MCH-trained flies: MCH (CS+)] in thirsty-state. (B) Quantification of the GCaMP6 responses to the training odor (CS+) relative to the non-training odor (CS−) in the α-lobe region of αβ core neurons 24-hour post-conditioning in OCT-trained (red bar) or MCH-trained (green bar) flies. The Log ratios of the CS+ response to the CS− response were calculated using the peak response amplitudes. Each value represents mean ± SEM (N = 6~11). Each bar is not statistically significantly different from zero, p > 0.05; one sample t-test. Genotype: *UAS-G-CaMP6m/+; VT0841-GAL4/+*.
(TIF)

**S5 Fig. Horizontal lobe of αβ surface neurons forms wLTM trace.** (A) The GCaMP6 response 24-hour after water-reward conditioning was assayed in αβ surface neurons (the image-recording region is showed in the top left figure). For the paired training group: flies received CS− odor without water-reward (US), followed by exposure to the CS+ odor with water-reward. For the unpaired training group: flies received CS− odor without water-reward, followed by exposure to CS+ odor without water-reward, and the water-reward was delivered 1-minute later after CS+ odor. Odor/water paired training induced an increase in the GCaMP6 responses in the β-lobe region of the αβ surface neurons to the training odor [OCT-trained flies: OCT (CS+), MCH-trained flies: MCH (CS+)] in thirsty-state. (B) Quantification of the increased GCaMP6 responses to the training odor (CS+) relative to the non-training odor (CS−) in the β-lobe region of αβ surface neurons 24-hour post-conditioning in OCT-trained (red bar) or MCH-trained (green bar) flies. The Log ratios of the CS+ response to the CS− response were calculated using the peak response amplitudes. Each value represents mean ± SEM (N = 6~8). *, p < 0.05; statistically significantly different from zero; one sample t-test. Genotype: *UAS-GCaMP6m/+; VT20803-GAL4/+*.
(TIF)

**S6 Fig. γ dorsal neurons show decreased calcium responses to odors.** (A) Naïve flies carrying *UAS-GCaMP6m/+; R93G04-GAL4/+* transgenes were used to perform calcium imaging experiment of odor response. Flies show significantly decreased calcium responses to OCT and MCH in each γ dorsal subdomain. (B) Quantification of the GCaMP6 responses to OCT and MCH in each γ dorsal subdomain in naïve flies. Each value represents mean ± SEM (N = 9).
(TIF)

**S7 Fig. γ main neurons do not form wLTM trace.** (A) The GCaMP6 response 24-hour after water-reward conditioning was assayed in γ main neurons (the image-recording region is showed in the top left figure). For the paired training group: flies received CS− odor without water-reward (US), followed by exposure to the CS+ odor with water-reward. For the unpaired training group: flies received CS− odor without water-reward, followed by exposure to CS+ odor without water-reward, and the water-reward was delivered 1-minute later after CS+ odor. Odor/water paired training did not induce wLTM trace 24-hour post-conditioning in the γ-lobe region of the γ main neurons to the training odor [OCT-trained flies: OCT (CS+), MCH-trained flies: MCH (CS+)] in thirsty-state. (B) Quantification of the GCaMP6 responses to the training odor (CS+) relative to the non-training odor (CS−) in the γ-lobe region of the γ main neurons in OCT-trained (red bar) or MCH-trained (green bar) flies 24-hour post-conditioning. The Log ratios of the CS+ response to the CS− response were calculated using the peak response amplitudes. Each value represents mean ± SEM (N = 7). Each bar is not statistically significantly different from zero, p > 0.05; one sample t-test. Genotype: *UAS-GCaMP6m/+; R64C08-GAL4/+*.
(TIF)

**S1 Table. Numerical data for graphs.**
(XLSX)

# Acknowledgments

We thank Wei-Huan Shyu for help with the initial setup of *in vivo* calcium imaging assays. We thank Bloomington *Drosophila* Stock Center, Vienna *Drosophila* RNAi Center, Vienna Tile (VT) Library, Fly Core in Taiwan, and Ann-Shyn Chiang for providing fly stocks.

## Author Contributions

**Conceptualization:** Wang-Pao Lee, Chia-Lin Wu.

**Data curation:** Wang-Pao Lee, Meng-Hsuan Chiang, Li-Yun Chang, Jhen-Yi Lee, Ya-Lun Tsai, Tai-Hsiang Chiu.

**Formal analysis:** Wang-Pao Lee, Meng-Hsuan Chiang, Li-Yun Chang, Jhen-Yi Lee, Ya-Lun Tsai, Tai-Hsiang Chiu, Chia-Lin Wu.

**Funding acquisition:** Chia-Lin Wu.

**Investigation:** Wang-Pao Lee, Meng-Hsuan Chiang, Li-Yun Chang, Jhen-Yi Lee, Ya-Lun Tsai, Tai-Hsiang Chiu, Chia-Lin Wu.

**Methodology:** Wang-Pao Lee, Meng-Hsuan Chiang, Hsueh-Cheng Chiang, Tsai-Feng Fu, Tony Wu, Chia-Lin Wu.

**Project administration:** Chia-Lin Wu.

**Resources:** Chia-Lin Wu.

**Supervision:** Chia-Lin Wu.

**Validation:** Wang-Pao Lee, Chia-Lin Wu.

**Visualization:** Wang-Pao Lee, Chia-Lin Wu.

**Writing – original draft:** Chia-Lin Wu.

**Writing – review & editing:** Wang-Pao Lee, Chia-Lin Wu.

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
