## [Decision Letter · Decision Letter 0]

5 May 2020

Dear Dr Wu,

Thank you very much for submitting your Research Article entitled 'Neural circuits encode CREB2-dependent water-reward long-term memory' to PLOS Genetics. Your manuscript was fully evaluated at the editorial level and by independent peer reviewers. The reviewers appreciated the attention to an important topic but identified some aspects of the manuscript that should be improved.

We therefore ask you to modify the manuscript according to the review recommendations before we can consider your manuscript for acceptance. Your revisions should address the specific points made by each reviewer.

[LINK]

Yours sincerely,

Gaiti Hasan

Associate Editor

PLOS Genetics

Gregory Barsh

Editor-in-Chief

PLOS Genetics

Reviewer's Responses to Questions

**Comments to the Authors:**

Reviewer #1: This work systematically investigates the location of water-reward long-term memory (wLTM) among different types of the mushroom body (MB) neurons in Drosophila. The data are very solid. The results are instructive for studying neural circuits of long-term memory. It is suitable to publish in PLOS Genetics. There are some minor concerns below.

1, In the abstract, “protein synthesis followed by the conserved cAMP response element binding protein (CREB)-dependent gene transcription”. The order of protein synthesis and transcription seems reversed.

2, Line 49, “… required for retrieval, but not for acquisition or consolidation of wLTM”. There is no evidence for such a statement in this manuscript. The previous study should be cited here.

3, In figure 2 and figure 5, it is more clear and helpful for readers if the information about neuron types (γ main, γ dorsal, or …) driven by different Gal4 lines is also provided in the figures.

4, In figure 6 and figure 7, it is better to provide a sketch of imaging protocol (time points for thirsty, training, imaging, and how to perform unpaired training).

Reviewer #2: In this manuscript, Lee et al characterize the role of the mushroom bodies in forming long-term water memories (wLTM). Drosophila have long been a leading model for the study of associative memory formation, historically relying on a paradigm where electric shock is paired with an odor. The study of water memories provide a number of significant advantages, including its ethological relevance and greater potential to map neural circuits carrying the US to the mushroom bodies. Therefore, studying wLTM is likely to be of broad interest to the community. This manuscript describes an acute requirement for protein synthesis in the mushroom bodies, and identifies physiological changes in defined lobes that are associated with memory. Overall, the manuscript is well written and scientifically sound. I have some concerns about whether the findings presented represent a significant advance over the groups 2017 paper that defined MB-neurons required for LTM formation. I have made a number of suggestions below that may help address potential weaknesses and enhance the impact of the manuscript. Overall, this manuscript is likely to be well received based on the under-studied nature of wLTM and the identification of physiological changes within specific MB lobes that associate with memory formation.

Major comments

1. 2.Based on the temperature shift protocols, it seems that expression dCREB2 and dRicin for 8 hrs after training impairs long-term memory. An alternative explanation is these manipulations induce irrevocable harm to the neurons. For these reasons the conclusion really requires a time-point the flies can recover from, and not be impacted (for example, 8-16hrs after training, or immediately prior to training).

2. For shibire experiments in figure 3, it would be helpful to also examine the consolidation period after testing, when protein-synthesis is found to be critical.

3. I have concerns about the use of the word engram throughout the manuscript. My understand is engram is the site of memory formation. Here, it seems this is interpreted more broadly and applied to any physiological change associated with memory.

4. Discussion does nto sufficiently address a few key aspects that will represent these findings in the broader context of fly memory. First, many types of fly memories are mushroom body dependent. At the level of neural circuits, what is shared and what is different? Second, are there further subsets of neurons within the MBs, and if so, how should this be interpreted given that the driver lines used for imaging cannot parse these subsets?

Minor Comments

1. For non-experts, it would be helpful to include images, or at least diagram showing the mushroom body lobes in Figures 1 and 2. Also please specify the expression pattern near the graph, in addition to the driver line.

Reviewer #3: Lee et al. addressed different subsets of mushroom body neurons for water-rewarded long-term memory (wLTM). The main findings of this work are: 1) post-training protein synthesis and dCREB2b functions in a/b-surface and g-dorsal neurons are essential for wLTM; 2) neurotransmission of the neurons at the wLTM retrieval is necessary for conditioned odor approach; 3) odor responses in these mushroom body neurons undergo training-dependent changes. Overall, results are more or less straightforward, and I am generally supportive for publication if the authors address following points.

1) Discussion is very minimalistic. For example, the Wu lab identified the PAM-b’1 neurons conveying water reward for wLTM (Shyu 2017 Nat Commun), while the current study showed that a’/b’ neurons have limited roles in wLTM. Why? How do the authors suggest the reward input in the b’ lobe causes associative plasticity in the a/b and g neurons? Another point of question is whether post-training protein synthesis is required in the a/b-surface and g-dorsal neurons. This could have been easily addressed with ricin-cs. The authors should provide more discussion.

2) “Engram neurons” and “cellular memory trace” should be used more carefully. These terms refer to cells that receive CS and US information and undergo plastic changes necessary and sufficient for memory (Gerber 2004 Curr Opin Neurobiol; Tonegawa 2015 Neuron). In that sense, the authors’ definition “The cellular memory traces can be any neural activity change induced by learning” (L173-4) is too forgiving. Under this definition, whatever changes downstream of neurons undergoing associative plasiticty can be included. The authors should tone down because these neurons undergoing training-dependent odor responses may not receive the information of water reward during training (cf. Shyu 2017).

3) Potential effects on water drinking by expression of dCREB2b in Kenyon cells should be experimentally tested because there are several studies showing that these cells are also important in regulation of feeding (e.g. Tsao 2018 eLife; Landayan 2018 Sci Rep).

4) I do not understand why the authors switched to use shibirets1 to narrow down the KC types (Figure 3, 4). Basically, expressions of shi and dCREB2b/ricin-cs address different cellular processes.

5) The title should be more specific. Current title just says that CREB2 is necessary for wLTM in some neural circuits.

6) Is there any evidence supporting the claim “wLTM can last for over 24-hours without any decay” (L24)? Shyu 2017 Nat Comm seems to show some, not huge though, decay.

7) “ANOVA followed by Tukey’s test” (694-5, 719-20). Is it correct? It looks the authors tested the difference from zero (Figure 6B, 7B).

**Have all data underlying the figures and results presented in the manuscript been provided?**

Reviewer #1: None

Reviewer #2: Yes

Reviewer #3: None

PLOS authors have the option to publish the peer review history of their article (what does this mean?). If published, this will include your full peer review and any attached files.

Reviewer #1: No

Reviewer #2: No

Reviewer #3: No

---

## [Decision Letter · Decision Letter 1]

29 Jun 2020

Dear Dr Wu,

We are pleased to inform you that your manuscript entitled "Mushroom body subsets encode CREB2-dependent water-reward long-term memory in Drosophila" has been editorially accepted for publication in PLOS Genetics. Congratulations!

Yours sincerely,

Gaiti Hasan

Associate Editor

PLOS Genetics

Gregory Barsh

Editor-in-Chief

PLOS Genetics

Comments from the reviewers (if applicable):

Reviewer's Responses to Questions

**Comments to the Authors:**

Reviewer #1: All my concerns have been cleared.

Reviewer #2: The authors have addressed all of my concerns. The revised manuscript includes additional justification for experiments and references that alleviate my concerns about multiple controls. In addition this version includes additional images and discussion that improve accessibility to general readership.

Reviewer #3: The authors addressed all my comments in this revision. Although some points could have been more completely (e.g. the term "memory trace" as pointed by another reviewer, too), these should be a subject of post-publication discussion. It is overall ready to be published.

**Have all data underlying the figures and results presented in the manuscript been provided?**

Reviewer #1: None

Reviewer #2: Yes

Reviewer #3: None

PLOS authors have the option to publish the peer review history of their article (what does this mean?). If published, this will include your full peer review and any attached files.

Reviewer #1: No

Reviewer #2: **Yes: **Alex Keene

Reviewer #3: No

**Data Deposition**

http://datadryad.org/submit?journalID=pgenetics&manu=PGENETICS-D-20-00452R1

**Press Queries**

---

## [Editor Report · Acceptance letter]

23 Jul 2020

PGENETICS-D-20-00452R1 

Mushroom body subsets encode CREB2-dependent water-reward long-term memory in Drosophila 

Dear Dr Wu, 

We are pleased to inform you that your manuscript entitled "Mushroom body subsets encode CREB2-dependent water-reward long-term memory in Drosophila" has been formally accepted for publication in PLOS Genetics! Your manuscript is now with our production department and you will be notified of the publication date in due course.

With kind regards,

Jason Norris

PLOS Genetics

On behalf of:
